# Cotton Fiber Strength Measurement and Its Relation to Structural Properties from Fourier Transform Infrared Spectroscopic Characterization

Yongliang Liu

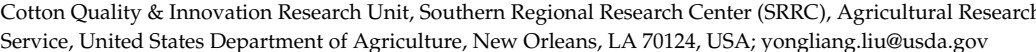

Cotton Quality & Innovation Research Unit, Southern Regional Research Center (SRRC), Agricultural Research Service, United States Department of Agriculture, New Orleans, LA 70124, USA; yongliang.liu@usda.gov

**Abstract:** There has been an interest in understanding the relationship between textile cotton fiber strength (or tenacity) and structure for better fiber quality measurement and enhancement. This study utilized coupled Stelometer and high volume instrument (HVI) measurements with attenuated total reflection Fourier transform infrared spectroscopy methods to relate fiber strength and associated properties (Stelometer elongation and HVI micronaire) with structure properties on six Upland (as A, B, C, D, E, and F) and one Pima cultivar. Although Stelometer tenacity agreed with HVI strength in general, the Upland D cultivar (immature) was observed to show the lowest HVI strength value, while the Upland F cultivar (larger infrared crystallinity index) was found to possess the smallest Stelometer tenacity value. A few strong and significant correlations were noted, for example, between infrared crystallinity and Stelometer elongation for the Upland A fibers, between infrared maturity and Stelometer tenacity for the Upland C fibers, and between infrared maturity and HVI strength for the Upland D fibers. Furthermore, there were apparent distinctions in regressions and statistics of examined correlations between each Upland cultivar and their combined fiber set, addressing the challenge of understanding the unique response between fiber physical and structure properties from different measurements even within one cotton cultivar.

**Keywords:** cotton fiber; fiber strength; fiber elongation; high volume instrument (HVI); Stelometer; fiber maturity; fiber crystallinity; Fourier transform infrared (FT-IR) spectroscopy





## 1. Introduction

Strength is one of the essential physical properties of textile cotton fibers. Its measurement can be categorized into a single fiber test or a bundle fiber test. The methods for a single fiber strength measurement involve the Mantis single-fiber tester [1,2], the Instron tensile tester [3], and the Favimat single-fiber tester [4,5], whereas the techniques for a bundle fiber strength measurement include the Stelometer bundle tester [6], Fibrotest [7], and the high volume instrument (HVI) [8]. Researchers have attempted to relate one strength test to another [2,4,7]. For example, Thibodeaux et al. [2] examined the relationship between the single-fiber strength from the Mantis test and bundle fiber strength from the Stelometer and HVI and observed that both the Stelometer and HVI bundle strength were linearly proportional to the ratio of the average Mantis breaking load to the square of the average fiber ribbon width. Delhom et al. [4] found that the single-fiber test produced higher mean values than the bundle test on the basis of the single-fiber test from the Favimat and bundle fiber measurements from the Stelometer and HVI. Cui et al. [7] studied the relationship between Fibrotest and HVI strengths on 12 cotton samples with different micronaire (MIC) values and noted an insignificant effect of the MIC value on the strength difference between the two measurements. Also, they observed low Fibrotest strengths compared to the Stelometer strength for the three international calibration cottons that had standard Stelometer strength values.

From the perspective of cotton research for better fiber quality measurement and enhancement, there is still an interest in how cotton fiber strength relates to fiber structure. Given the complexity of both fiber strength measurement and structure determination, researchers have taken different strategies. For example, Benedict et al. [9] reported a correlation of 0.94 between the average length of the cellulose chains in the crystalline cellulose and the HVI bundle fiber strength after analyzing the crystalline microfibrillar fragments by $^{13}$C-nuclear magnetic resonance (NMR) spectroscopy. Hsieh's group [1,3,10] developed the "XRAY" X-ray diffraction (XRD) analysis program to assess the crystallite size and crystallinity parameters of dried developing cotton fibers from the wide-angle XRD pattern and further correlated the crystallite size and crystallinity with fiber-breaking force and tenacity (or strength) that were measured from a Mantis single fiber tester or an Instron tensile tester. The results indicated that the single fiber breaking forces were positively related to both crystallite size and crystallinity, and the breaking force and tenacity increase appeared to be related more to crystallite size than to crystallinity. In addition, the pattern of single-fiber breaking tenacities against fiber crystallinity differed between cotton fibers from two genotypes of SJ-2 and Maxxa. They found that other structural parameters than the crystallite sizes and crystallinity, such as fibril orientation and residual stress, may play important roles in impacting the single fiber strength of cotton fibers. Using a different approach, Liu et al. [11] estimated the fiber infrared maturity ($M_{IR}$) and infrared crystallinity ($CI_{IR}$) of Upland and Pima cotton fibers from attenuated total reflection Fourier transform infrared (ATR FT-IR) spectroscopy in an effort to relate both $M_{IR}$ and $CI_{IR}$ values to Stelometer tenacity ($STE_{ten}$) and elongation ($STE_{elo}$) properties. The uniqueness of this work was the capability of the ATR FT-IR method to scan tiny Stelometer breakage specimens (2~5 mg), which cannot be readily analyzed by a conventional XRD measurement or by chemical extraction and identification analysis. Compared to an increase in fiber $STE_{ten}$ with either $M_{IR}$ or $CI_{IR}$ for Pima fibers, there was an unclear trend between the two for the combined Upland fiber set from the U.S. (two varieties) and outside of the U.S. (14 varieties from 2 Asian and 2 African countries). Based on comprehensive measurements from HVI, cross-sectional image analysis, Cottonscope, and Favimat, Kim et al. [12] investigated the effect of fiber maturity on the bundle and single fiber strength of Upland cotton constructed from a genetic approach. They showed significant and positive correlations between MIC and HVI bundle fiber strength or elongation value, and also between MIC or maturity ratio value and single fiber breaking force rather than with single fiber strength. Overall, making a direct comparison between these studies is challenging because of differences in bundle/single fiber strength measurement methods (Stelometer, HVI, Instron tensile tester, and Favimat), fiber crystallinity/maturity determination methods (XRD, $^{13}$C NMR, ATR FT-IR, HVI, cross-sectional image analysis, and Cottonscope), and fiber sources (field-grown or greenhouse-grown).

Both fiber $STE_{ten}$ and HVI strength ($HVI_{str}$) reflect the external force-induced breaking of cotton bundle fibers, but the values have been calculated differently. Unlike the Stelometer test, which uses the weight of a broken bundle beard to normalize the breaking force for assessing the $STE_{ten}$, the HVI test uses a fiber MIC as the mass substitute for $HVI_{str}$. Other factors such as fiber sample size, fiber bundle orientation, pre-tension, speed of the break, and clamp placement influence the $STE_{ten}$ and $HVI_{str}$ values [13]. In an earlier investigation, Liu et al. [11] examined the relationship between the $STE_{ten}$ or $STE_{elo}$ property and the $M_{IR}$ or $CI_{IR}$ index on a combined Upland fiber set, including two U.S. varieties. Taking a similar approach, this study related $STE_{ten}$ and $STE_{elo}$ properties as well as HVI MIC and $HVI_{str}$ properties to $M_{IR}$ or $CI_{IR}$ values on six U.S. Upland cultivars. The main objectives of this study were: (1) to evaluate the bundle strength agreement between $HVI_{str}$ and $STE_{ten}$ measurements on six Upland cotton cultivars and (2) to correlate the $STE_{ten}$ and $STE_{elo}$ properties as well as HVI MIC and $HVI_{str}$ properties with $M_{IR}$ and $CI_{IR}$ indices from ATR FT-IR measurements on Stelometer fiber breakage specimens of these fibers. As a comparison to Upland cultivars, Pima cottons were included in the analysis.

## 2. Materials and Methods

### 2.1. Cotton Lint Samples

A total of 31 Upland lot samples representing 6 Upland cotton cultivars and 18 Pima fiber samples from 1 Pima cultivar were collected randomly. These cotton fibers were grown in the U.S., but their specific geographic origins and crop years were not available. There were 4, 5, 6, 5, 3, and 8 fiber lots in respective Upland A, B, C, D, E, and F cultivars, and samples in a cultivar were replicated field lots from the same location. They were well conditioned at a constant relative humidity of 65 ± 2% and a temperature of 21 ± 1 °C for at least 48 h prior to routine HVI and Stelometer testing as well as the ATR FT-IR spectral acquisition.

### 2.2. Fiber HVI and Stelometer Property Measurement

Average $HVI_{str}$ and MIC properties were obtained from five replicates on each sample by an Uster HVI 900A system (Zellweger Uster Inc., Knoxville, TN, USA). Mean $STE_{ten}$ and $STE_{elo}$ values were determined from three replicates on individual samples by the use of a Stelometer flat bundle tester (Spinlab, Knoxville, TN, USA) with 1/8 inch (3.2 mm) clamp spacing as detailed previously [11]. All broken bundles free of any impurities (or non-lint materials) were retained for subsequent ATR FT-IR spectral scans.

### 2.3. Fiber $M_{IR}$ and $CI_{IR}$ Calculation

Fiber $M_{IR}$ and $CI_{IR}$ values were estimated by applying simple algorithmic analysis to the ATR FT-IR spectra, as shown in Figure 1, with the proposed procedure [14]. Briefly, after the spectra were exported into Microsoft Excel, the first algorithmic $R_1$ equation below was used to calculate the $R_1$ value:

$$R_1 = (I_{956} - I_{1500})/(I_{1032} - I_{1500}) \tag{1}$$

where $I_{1500}$, $I_{1032}$, and $I_{956}$ are each a three-point intensity average at respective wavenumbers. The $I_{1032}$ characterized the large and positive intensity variation, while the $I_{956}$ represented the large and negative intensity variation in the difference spectrum between immature and mature cotton. The $I_{1500}$ was used to offset two readings because of its minimum absorbance.

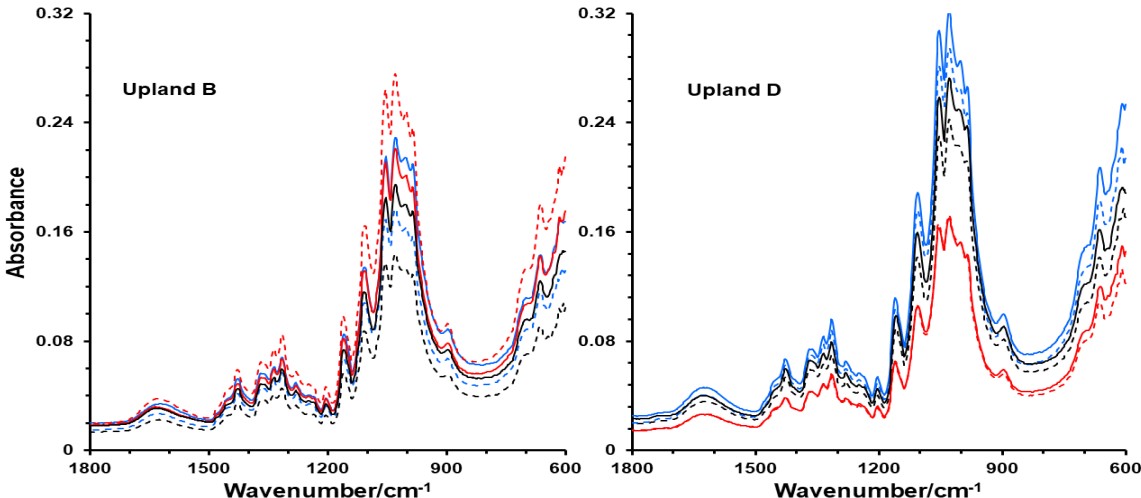

**Figure 1.** Representative six raw ATR FT-IR spectra (2 spectra/bundle shown in solid and dotted lines × 3 bundles given in black, red, and blue colors) in the 1800–600 cm$^{-1}$ region from one Upland B sample (MIC = 4.70) or one Upland D sample (MIC = 2.96). $M_{IR}$ or $CI_{IR}$ values for each sample were the averages of $M_{IR}$ or $CI_{IR}$ values from six raw ATR FT-IR spectral measurements.

Next, the second algorithm, $M_{IR}$, was used to convert the $R_1$ value into the $M_{IR}$ index:

$$M_{IR} = (R_1 - R_{1,sm})/(R_{1,lr} - R_{1,sm}) \tag{2}$$

in which $R_1$, $R_{1,lr}$, and $R_{1,sm}$ are the $R_1$ values for the unknown sample, the largest $R_1$, and the smallest $R_1$, respectively. The $R_{1,sm}$ and $R_{1,lr}$ values were determined to be 0.14 and 0.59.

Similarly, $CI_{IR}$ was computed using two algorithms, with the first algorithm $R_2$ utilizing three respective IR intensities at 800, 730, and 708 cm$^{-1}$, and the second algorithm $CI_{IR}$ (%) changing the $R_2$ values into fiber $CI_{IR}$.

$$R_2 = (I_{708} - I_{800})/(I_{730} - I_{800}) \tag{3}$$

$$CI_{IR} \text{ (\%)} = ((R_2 - 1.4)/2.0) \times 100 \tag{4}$$

To collect the spectra, an FTS 3000MX Fourier transform IR spectrometer (Varian Instruments, Randolph, MA, USA) equipped with a ceramic source, KBr beam splitter, deuterated triglycine sulfate (DTGS) detector, and an ATR attachment was used. Two spectra in the absorbance unit were collected for each broken bundle over the range of 4000–600 cm$^{-1}$ at 4 cm$^{-1}$ with 32 co-added scans, and the mean spectra for each sample were exported into Microsoft Excel 2016 to assess fiber $CI_{IR}$ and $M_{IR}$ indices.

The sampling depth of the ATR device is from 2 to 15 μm, depending on the ATR crystal materials and wavenumber range [15,16], while the thickness of the secondary cell wall (SCW) in mature cotton fibers varies from 2 to 7 μm [17]. Therefore, the ATR FT-IR method is capable of representing the information inside mature cotton fibers by the use of both a low refractive index crystal (ZnSe or diamond) and a low spectral region (1100 to 600 cm$^{-1}$). Overall, the use of the ATR device greatly facilitates the sampling procedure and is also time-efficient for a large number of sample analyses.

### 2.4. Mathematical and Statistical Analysis

Mathematical and statistical analyses were conducted using Microsoft Excel 2016. For mathematical analysis, Pearson correlation coefficient (*r*), slope, and coefficient of determination (R$^2$) were acquired from the relationships among fiber Stelometer, HVI, and spectral values by fitting the data into a linear regression trendline option. When the $|r|$ value is less than 0.3, between 0.3 and 0.7, or greater than 0.7, a weak, moderate, or strong (either positive or negative) linear correlation exists between the two variables [18]. For statistical analysis, *p*-values among a pair of fibers' physical and structural properties were calculated using the Excel regression function under Data Analysis. The *p*-value cutoff for significance was 0.05, with three levels at 0.05~0.01 (*), 0.01~0.001 (**), and <0.001 (***).

## 3. Results and Discussion

### 3.1. Fiber STE$_{ten}$ vs. STE$_{elo}$ from Stelometer Measurement

The relationships between fiber STE$_{ten}$ and STE$_{elo}$ for Upland and Pima cultivars are depicted in Figure 2. As expected, the Pima cultivar exhibited a significantly higher STE$_{ten}$ (24.3–29.3 g/tex) than the six Upland cultivars (18.4–26.2 g/tex) (*p*-value < 0.001), and the six Upland cultivars showed apparent differences in STE$_{ten}$ among them. Average STE$_{ten}$ values summarized in Table 1 for the Pima cultivar as well as Upland A, B, C, D, E, and F cultivars were 27.3, 24.1, 22.2, 21.5, 21.3, 20.9, and 19.9 g/tex, respectively. Meanwhile, the Pima cultivar showed a statistical difference in STE$_{elo}$ from six Upland cultivars (*p*-value = 0.02), and six Upland cultivars showed a little difference among them (Figure 2). Respective mean STE$_{elo}$ values were 7.0, 6.2, 6.3, 7.2, 6.5, 6.6, and 6.5% for the Pima cultivar as well as Upland A, B, C, D, E, and F cultivars (Table 1).

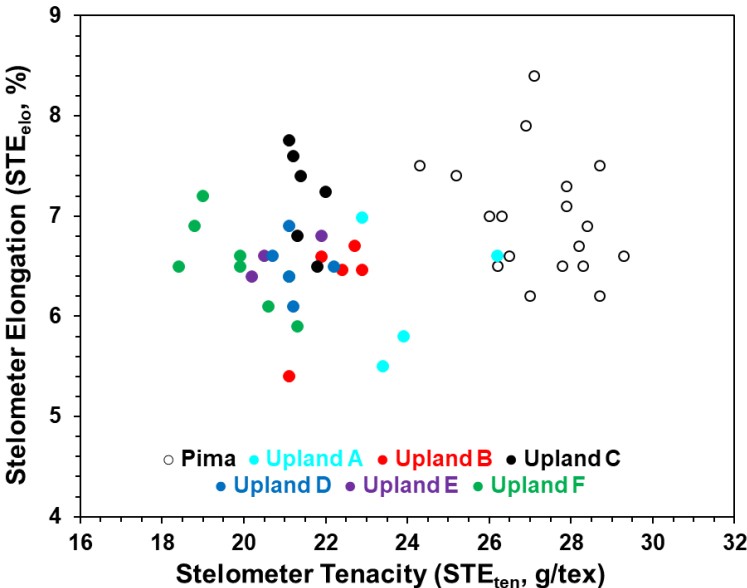

**Figure 2.** Plot of STE$_{elo}$ vs. STE$_{ten}$ for six Upland cultivars (A, B, C, D, E, and F) and one Pima cultivar.

**Table 1.** Comparison of mean and range of STE$_{ten}$, STE$_{elo}$, HVI$_{str}$, HVI MIC, $M_{IR}$, and $CI_{IR}$ values for six Upland and one Pima cultivar.

| Cultivar | STE$_{ten}$ | | STE$_{elo}$ (%) | | HVI$_{str}$ | | HVI MIC | | $M_{IR}$ | | $CI_{IR}$ (%) | |
|---|---|---|---|---|---|---|---|---|---|---|---|---|
| | Mean | Range | Mean | Range | Mean | Range | Mean | Range | Mean | Range | Mean | Range |
| Upland A | 24.1 | 22.9~26.2 | 6.2 | 5.5~7.0 | 34.6 | 33.5~35.6 | 4.15 | 4.04~4.36 | 0.74 | 0.68~0.78 | 66.4 | 64.2~68.5 |
| Upland B | 22.2 | 21.0~22.9 | 6.3 | 5.4~6.7 | 33.8 | 33.2~34.4 | 4.77 | 4.59~4.96 | 0.79 | 0.75~0.83 | 69.6 | 65.2~75.1 |
| Upland C | 21.5 | 21.1~22.0 | 7.2 | 6.5~7.8 | 28.8 | 27.2~30.0 | 4.42 | 4.23~4.53 | 0.75 | 0.69~0.81 | 61.4 | 57.4~65.4 |
| Upland D | 21.3 | 20.7~22.2 | 6.5 | 6.1~6.9 | 25.7 | 24.2~27.3 | 3.03 | 2.96~3.10 | 0.64 | 0.59~0.68 | 58.4 | 52.8~63.5 |
| Upland E | 20.9 | 20.2~21.9 | 6.6 | 6.4~6.8 | 30.1 | 29.5~30.9 | 4.52 | 4.44~4.60 | 0.80 | 0.76~0.83 | 65.5 | 63.7~67.5 |
| Upland F | 19.9 | 18.4~21.3 | 6.5 | 5.9~7.2 | 29.5 | 28.1~31.1 | 4.68 | 4.52~4.94 | 0.79 | 0.72~0.85 | 83.1 | 74.1~95.8 |
| Pima | 27.3 | 24.3~29.3 | 7.0 | 6.2~8.4 | 36.3 | 33.3~41.2 | 3.90 | 3.73~4.15 | 0.77 | 0.71~0.84 | 72.6 | 62.6~84.8 |

As summarized in Table 2, fiber elongation was correlated with tenacity weakly for all Upland datasets ($r = -0.17$) or Pima fibers ($r = -0.30$). Throughout this work, a negative *r* value means that as one variable increases, the other decreases in the dataset, and vice versa. Despite a limited number of fiber samples within each Upland cultivar, Table 2 revealed a complicated scenario between STE$_{elo}$ and STE$_{ten}$ among six Upland cultivars. Three Upland cultivars (C, D, and F) showed an increase in STE$_{ten}$ with decreasing STE$_{elo}$ ($r = -0.74$ to $-0.14$), whereas the other three cultivars (A, B, and E) were in an opposite pattern ($r = 0.14$ to $0.94$). Only Upland F fibers showed a strong and significant negative relationship between STE$_{elo}$ and STE$_{ten}$ ($r = -0.74$, *p*-value = 0.04). The difference in genotype and growth environment and their interactions within Upland or Pima cultivars are probably responsible for the observation in Table 2, and more studies might be necessary to look into why cultivars acted differently when relating fiber tenacity to elongation as well as other properties discussed below.

**Table 2.** Comparison of *r* and significance (significant *, *p*-value = 0.05~0.01) between STE$_{elo}$ and STE$_{ten}$ for individual Upland cultivars, all Upland datasets, and one Pima cultivar.

| Cultivar | Upland | | | | | | | Pima |
|---|---|---|---|---|---|---|---|---|
| | A | B | C | D | E | F | All | |
| STE$_{elo}$ vs. STE$_{ten}$ | 0.14 | 0.82 | −0.55 | −0.14 | 0.94 | −0.74 * | −0.17 | −0.30 |

### 3.2. Fiber HVI_{str} vs. MIC from HVI Measurement

Similar to STE_ten in Figure 2, the Pima cultivar in Figure 3 showed a greater HVI_str than six Upland cultivars (*p*-value < 0.001), and six Upland cultivars had obvious differences in HVI_str within them. Average HVI_str values for the Pima cultivar as well as Upland A, B, C, D, E, and F cultivars were 36.3, 34.6, 33.8, 28.8, 25.7, 30.1, and 29.5 g/tex, respectively (Table 1). There was an unclear separation in MIC between the Pima cultivar and six Upland cultivars, but there were some differences in MIC among the six Upland cultivars. Average MIC values for the Pima cultivar as well as Upland A, B, C, D, E, and F cultivars were 3.90, 4.15, 4.77, 4.42, 3.03, 4.52, and 4.68, respectively (Table 1). Clearly, Upland D fibers were immature due to their quite low MIC (=3.03) as well as $M_{IR}$ (=0.64) and $M_{IR}$ (=58.4%) in Table 1. HVI elongation property on these fibers was measured with HVI_str and MIC simultaneously at the time. Since the HVI elongation measurement was not calibrated at that time, this study neither discussed HVI elongations nor compared them to STE_elo. Recently, McCormick et al. [19] developed cotton elongation standards from two commercial bales (with low and high HVI elongation) and used these potential standards to correct HVI against Stelometer elongation measurements. Their result showed that the corrected HVI elongations were at least as good as the Stelometer elongations.

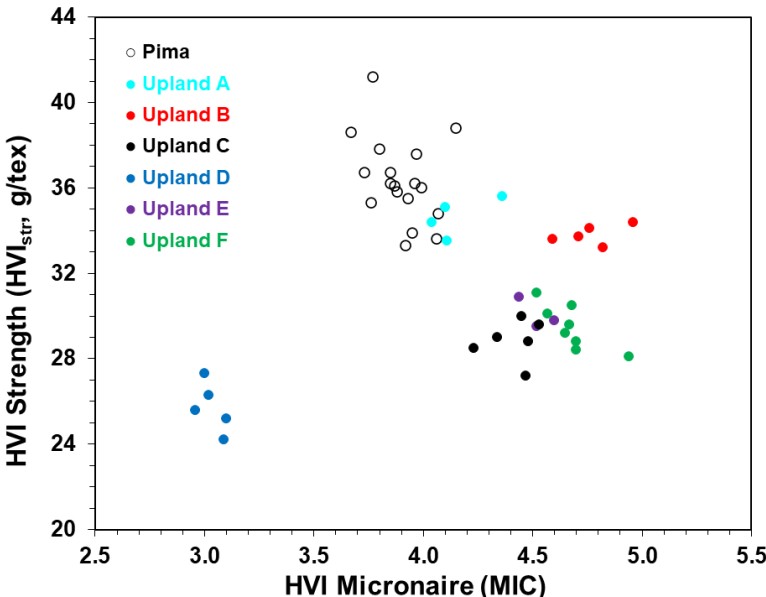

**Figure 3.** Plot of HVI_str vs. MIC for six Upland cultivars (A, B, C, D, E, F) and one Pima cultivar.

Table 3 suggested that, unlike Pima fibers that had a tendency to decrease in fiber HVI_str with elevating MIC insignificantly (*r* = −0.35, *p*-value > 0.05), all Upland datasets showed an increase in fiber HVI_str with MIC moderately and significantly (*r* = 0.55, *p*-value = 0.001). This observation resembled a significant but weak correlation between MIC and HVI_str on 168 Upland kinds of cotton, covering a broad range of maturity and strength [12]. Further examination indicated that three Upland cultivars (A, B, and C) increased in fiber HVI_str along with MIC (*r* = 0.49 to 0.65), whereas the remaining three cultivars (D, E, and F) decreased in fiber HVI_str with MIC (*r* = −0.79 to −0.62). Among them, only Upland F fibers showed a strong and significant negative relationship between fiber HVI_str and MIC (*r* = −0.79, *p*-value = 0.02), likely due to a higher $CI_{IR}$ index for Upland F fibers (Table 1). Hence, caution should be taken when compiling two or more Upland cultivars into one dataset to study the specific relationships as given in Table 2. This caution might be applied to the same cultivar in different geographic locations and crop years.

**Table 3.** Comparison of *r* and significance (significant *, *p*-value = 0.05~0.01; very significant **, *p*-value = 0.01~0.001) between $HVI_{str}$ and MIC for individual Upland cultivars, all Upland datasets, and one Pima cultivar.

| Cultivar | Upland | | | | | | | Pima |
|---|---|---|---|---|---|---|---|---|
| | A | B | C | D | E | F | All | |
| $HVI_{str}$ vs. MIC | 0.65 | 0.49 | 0.55 | −0.62 | −0.75 | −0.79 * | 0.55 ** | −0.35 |

*3.3. Fiber $M_{IR}$ vs. $CI_{IR}$ Index from ATR FT-IR Measurement*

Relating $CI_{IR}$ to $M_{IR}$ in Figure 4 and also the statistics in Table 4 showed an increase in fiber $CI_{IR}$ with $M_{IR}$ moderately to greatly and significantly for both all Upland datasets (*r* = 0.69, *p*-value < 0.001) and Pima fibers (*r* = 0.92, *p*-value < 0.001). Apparently, there were differences in synchronous fiber $M_{IR}$ and $CI_{IR}$ developments between Upland and Pima cultivars, as well as among six Upland cultivars. For example, more fiber $CI_{IR}$ production than $M_{IR}$ accumulation was observed for Upland F fibers (regression slope = 181.3), followed by Upland D fibers (slope = 111.6), Upland B fibers (slope = 99.1), Upland C fibers (slope = 49.0), Upland E fibers (slope = 25.3), and Upland A fibers (slope = 17.2). Upland D and F cultivars showed strong and significant relationships between fiber $CI_{IR}$ and $M_{IR}$ (*r* = 0.87 to 0.92, *p*-value = 0.03).

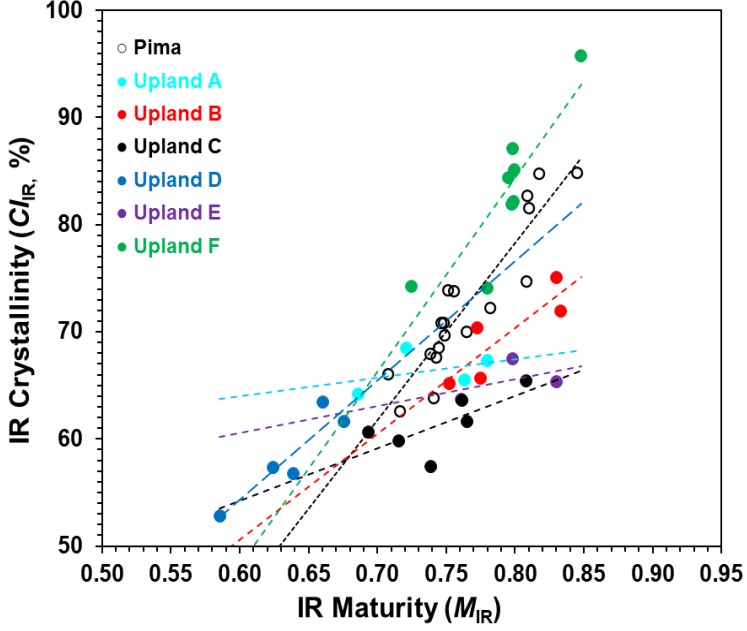

**Figure 4.** Plot of $CI_{IR}$ vs. $M_{IR}$ for six Upland cultivars (A, B, C, D, E, and F) and one Pima cultivar.

**Table 4.** Comparison of *r* and significance (significant *, *p*-value = 0.05~0.01; very significant *** at *p* < 0.001, *p* < 0.001) between $CI_{IR}$ and $M_{IR}$ for individual Upland cultivars, all Upland datasets, and one Pima cultivar.

| Cultivar | Upland | | | | | | | Pima |
|---|---|---|---|---|---|---|---|---|
| | A | B | C | D | E | F | All | |
| $CI_{IR}$ vs. $M_{IR}$ | 0.39 | 0.86 | 0.71 | 0.92 * | 0.46 | 0.87 * | 0.69 *** | 0.92 *** |

Relative to Upland D fibers in Table 1 that had the smallest $M_{IR}$ index (0.64 on average) and $CI_{IR}$ index (58.4%), other Upland cultivars showed larger $M_{IR}$ values (0.74 to 0.80) with varying $CI_{IR}$ readings (61.4 to 83.1%). Obviously, the $M_{IR}$ and $CI_{IR}$ values of Pima

fibers ($M_{IR}$ = 0.77 and $CI_{IR}$ = 72.6%) were within those of the Upland cultivars. The observation suggested that the ATR FT-IR spectral measurement could be applied to compare cotton fiber chemical and structural differences induced by cotton varieties and growth environments, but it could not be used to distinguish the Pima fibers from the Upland ones on the basis of $M_{IR}$ and $CI_{IR}$ values.

### 3.4. Relationship between Fiber $STE_{ten}$ and $HVI_{str}$

The plot of $HVI_{str}$ against $STE_{ten}$ in Figure 5 revealed a reasonable agreement between two measurements but with a scattered pattern. This indicated that the fiber strength testing mechanism is complicated and the measurement could be influenced by a number of factors, such as fiber elongation, elastic properties, cross-sectional area, length, crystallite size, fibril orientation, residual stress, strength uniformity, as well as the operator's experience [3,11,13, 19,20]. $STE_{ten}$ decreased in the order of Pima (27.3), Upland A (24.1), Upland B (22.2), Upland C (21.5), Upland D (21.3), Upland E (20.9), and Upland F (19.9). In contrast, $HVI_{str}$ decreased in the sequence of Pima (36.3), Upland A (34.6), Upland B (33.8), Upland E (30.1), Upland F (29.5), Upland C (28.8), and Upland D (25.7). Notably, the Upland D cultivar, immature with the smallest MIC, $M_{IR}$, and $CI_{IR}$ values among six Upland cultivars, showed the lowest $HVI_{str}$ value (25.7) but relatively larger $STE_{ten}$ value (21.3), while the Upland F cultivar, having the greater $CI_{IR}$ index than other Upland cultivars, showed the smallest $STE_{ten}$ value (19.9) but relatively larger $HVI_{str}$ value (29.5).

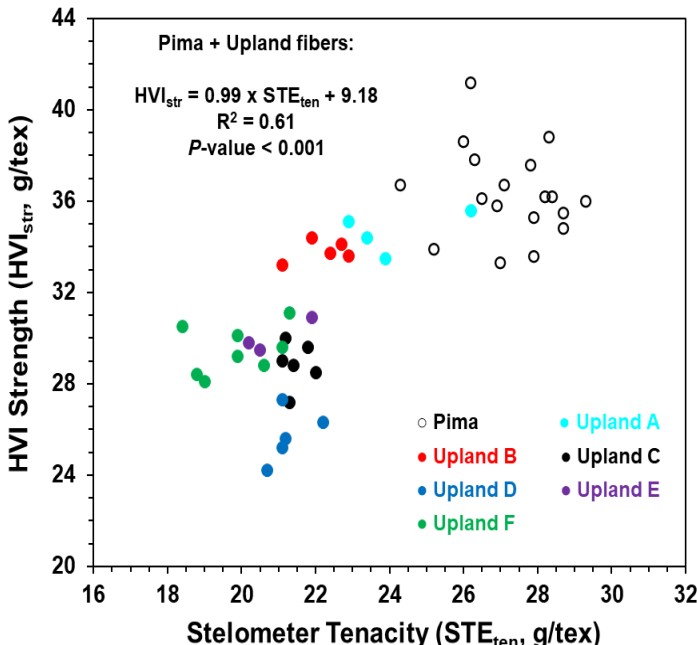

**Figure 5.** Plot of $HVI_{str}$ vs. $STE_{ten}$ for six Upland cultivars (A, B, C, D, E, and F) and one Pima cultivar.

Analysis of the pattern in Figure 5 suggested that (i) the Pima cultivar showed a clear cut in $STE_{ten}$ values from all but one Upland A sample ($p$-value < 0.001), whereas five Pima samples were overlapped in $HVI_{str}$ with two Upland cultivars, although the difference (Pima vs. Upland A and B) was significant ($p$-value = 0.004); (ii) Upland A cultivar differed in $STE_{ten}$ from the Upland B cultivar ($p$-value = 0.037), but their $HVI_{str}$ values were not statistically different ($p$-value > 0.05); (iii) three Upland cultivars (C, E, and F) showed a clear difference in $STE_{ten}$ ($p$-value = 0.003); however, they had an insignificant difference in $HVI_{str}$ ($p$-value > 0.05); and (iv) four Upland cultivars (B, C, D, and E) exhibited an insignificant difference in $STE_{ten}$ values ($p$-value > 0.05), but they showed significant differences in $HVI_{str}$ values ($p$-value < 0.001).

In general, there was a strong and significant correlation between $STE_{ten}$ and $HVI_{str}$ for combined Pima and Upland fibers ($r$ = 0.78, $p$-value < 0.001) in Figure 5, and also a

moderate and significant correlation for all Upland datasets ($r = 0.57$, *p*-value < 0.001) in Table 5. Opposed to the Upland C fibers and Pima fibers that showed a weak correlation between $HVI_{str}$ and $TE_{ten}$ ($r = -0.17$ to $-0.01$) in Table 5, five of six Upland cultivars (A, B, D, E, and F) indicated a moderate to strong correlation insignificantly ($r = 0.35$ to $0.93$, *p*-value > 0.05).

**Table 5.** Comparison of *r* and significance (very significant *** at $p < 0.001$, $p < 0.001$) between $STE_{ten}$ and $HVI_{str}$ for individual Upland cultivars, all Upland datasets, and one Pima cultivar.

| Cultivar | Upland | | | | | | | Pima |
|---|---|---|---|---|---|---|---|---|
| | A | B | C | D | E | F | All | |
| $HVI_{str}$ vs. $STE_{ten}$ | 0.47 | 0.35 | −0.01 | 0.47 | 0.93 | 0.36 | 0.57 *** | −0.17 |

In the aspect of consistent and good agreement between two strength measurements, the six Upland cultivars were grouped into three fiber sets according to mean $M_{IR}$ values. The first set consisted of three Upland cultivars (B, E, and F) with respective mean $M_{IR}$ values of 0.79, 0.80, and 0.79 (or mean MIC values of 4.77, 4.52, and 4.68); the second set included two Upland cultivars (A and C) with respective mean $M_{IR}$ values of 0.74 and 0.75 (or mean MIC values of 4.15 and 4.42); and the third set consisted of one Upland cultivar (D) with a mean $M_{IR}$ value of 0.64 (or a mean MIC value of 3.03). As anticipated, there were improved correlations between $STE_{ten}$ and $HVI_{str}$ for the first set ($r = 0.79$) and the second set ($r = 0.84$), compared to a *r* value of 0.57 for all Upland datasets in Table 5.

*3.5. Relationships between Fiber $M_{IR}$ or $CI_{IR}$ and MIC or $HVI_{str}$*

For all Upland datasets in Table 6, there was a stronger correlation between $M_{IR}$ and MIC ($r = 0.81$, *p*-value < 0.001) than between $CI_{IR}$ and MIC ($r = 0.57$, *p*-value = 0.001). It was consistent with a previous report that the MIC values of 16 crossed-Upland kinds of cotton were significantly correlated with fiber maturity and linear densities determined by Cottonscope and image analysis methods [12]. Further analysis revealed that MIC increased insignificantly with fiber $M_{IR}$ for Upland A and D fibers ($r = 0.66$ to $0.80$) and decreased with fiber $M_{IR}$ for the remaining four Upland cultivars ($r = -0.99$ to $-0.26$), in which only Upland E fibers possessed a strong and significant correlation ($r = -0.99$, *p*-value = 0.03). Similarly, MIC increased with fiber $CI_{IR}$ for Upland A and D cultivars, but none of the six Upland cultivars showed any significant correlations between fiber $CI_{IR}$ and MIC. For Pima fibers, there existed weak and insignificant correlations either between $M_{IR}$ and MIC ($r = 0.10$, *p*-value > 0.05) or between $CI_{IR}$ and MIC ($r = 0.10$, *p*-value > 0.05).

**Table 6.** Comparison of *r* and significance (significant *, *p*-value = 0.05~0.01; very significant **, *p*-value = 0.01~0.001; very significant *** at $p < 0.001$, $p < 0.001$) between $M_{IR}$ or $CI_{IR}$ and MIC or $HVI_{str}$ for individual Upland cultivars, all Upland datasets, and one Pima cultivar.

| Cultivar | Upland | | | | | | | Pima |
|---|---|---|---|---|---|---|---|---|
| | A | B | C | D | E | F | All | |
| $M_{IR}$ vs. MIC | 0.80 | −0.41 | −0.26 | 0.66 | −0.99 * | −0.57 | 0.81 *** | 0.10 |
| $CI_{IR}$ vs. MIC | 0.80 | −0.20 | −0.55 | 0.83 | −0.41 | −0.26 | 0.57 ** | 0.10 |
| $M_{IR}$ vs. $HVI_{str}$ | 0.14 | 0.01 | −0.39 | −0.98 ** | 0.71 | 0.55 | 0.46 ** | 0.10 |
| $CI_{IR}$ vs. $HVI_{str}$ | 0.66 | −0.28 | −0.49 | −0.85 | −0.30 | 0.46 | 0.20 | 0.26 |

Compared to a weak correlation between fiber $CI_{IR}$ and $HVI_{str}$ for all Upland datasets ($r = 0.20$, *p*-value > 0.05) in Table 6, there was a moderate and significant relationship between fiber $M_{IR}$ and $HVI_{str}$ ($r = 0.46$, *p*-value = 0.008). Four Upland cultivars (A, B, E, and F) showed an increase in $HVI_{str}$ with $M_{IR}$ ($r = 0.01$ to $0.71$), and the other two cultivars (C

and D) indicated a decrease in $HVI_{str}$ with $M_{IR}$ ($r = -0.98$ to $-0.39$), of which only Upland D fibers (immature) had a strong and significant correlation ($r = -0.98$, $p$-value = 0.002). Two of the six Upland cultivars (A and F) suggested an $HVI_{str}$ increase with fiber $CI_{IR}$ ($r = 0.46$ to 0.66); however, none of the six Upland cultivars showed any significant correlation between fiber $CI_{IR}$ and $HVI_{str}$. Within Pima fibers, weak and insignificant correlations were observed between either $M_{IR}$ and $HVI_{str}$ ($r = 0.10$, $p$-value > 0.05) or $CI_{IR}$ and $HVI_{str}$ ($r = 0.26$, $p$-value > 0.05).

### 3.6. Relationships between Fiber $M_{IR}$ or $CI_{IR}$ and $STE_{ten}$ or $STE_{elo}$

Contrary to a moderate and significant positive relationship between fiber $M_{IR}$ and $HVI_{str}$ ($r = 0.46$, $p$-value = 0.008) or a weak and insignificant positive correlation between fiber $CI_{IR}$ and $HVI_{str}$ ($r = 0.20$, $p$-value > 0.05) for all Upland datasets in Table 6, Table 7 implied a weak and insignificant negative correlation between fiber $M_{IR}$ and $STE_{ten}$ ($r = -0.01$, $p$-value > 0.05) or a negative moderate but significant correlation between fiber $CI_{IR}$ and $STE_{ten}$ ($r = -0.40$, $p$-value = 0.02). The differing $STE_{ten}$ and $HVI_{str}$ responses to $M_{IR}$ or $CI_{IR}$ indices could be due to differences in the way of normalizing the breaking force, as the Stelometer test uses the weight of a broken bundle while the HVI test uses a MIC. MIC was positively and significantly related to fiber maturity, represented by $M_{IR}$ for all Upland datasets in general (Table 6), or by fiber maturity and linear density reported earlier [12].

**Table 7.** Comparison of *r* and significance (significant *, $p$-value = 0.05~0.01) between $M_{IR}$ or $CI_{IR}$ and $STE_{ten}$ or $STE_{elo}$ for individual Upland cultivars, all Upland datasets, and one Pima cultivar.

| Cultivar | Upland | | | | | | | Pima |
| --- | --- | --- | --- | --- | --- | --- | --- | --- |
| | A | B | C | D | E | F | All | |
| $M_{IR}$ vs. $STE_{ten}$ | 0.75 | 0.47 | 0.90 * | −0.37 | 0.92 | −0.01 | −0.01 | 0.45 |
| $CI_{IR}$ vs. $STE_{ten}$ | 0.66 | 0.01 | 0.56 | −0.28 | 0.01 | −0.10 | −0.40 * | 0.58 * |
| $M_{IR}$ vs. $STE_{elo}$ | 0.35 | 0.33 | −0.62 | −0.49 | 0.99 * | −0.50 | −0.01 | 0.10 |
| $CI_{IR}$ vs. $STE_{elo}$ | 0.99 * | −0.10 | −0.40 | −0.36 | 0.41 | −0.35 | −0.22 | −0.14 |

As shown in Table 7, four Upland cultivars (A, B, C, and E) suggested an $STE_{ten}$ increase with $M_{IR}$ ($r = 0.47$ to 0.92), while the other two cultivars (D and F) indicated an opposite direction ($r = -0.37$ to $-0.01$), but only Upland C fibers had a strong and significant correlation ($r = 0.90$, $p$-value = 0.01). The same trend was observed for these six Upland cultivars when relating $STE_{ten}$ to fiber $CI_{IR}$; however, none of them showed significant correlations between fiber $CI_{IR}$ and $STE_{ten}$. Relative to a moderate and insignificant correlation between $M_{IR}$ and $STE_{ten}$ ($r = 0.45$, $p$-value > 0.05) for Pima fibers, there existed a moderate and significant correlation between $CI_{IR}$ and $STE_{ten}$ ($r = 0.58$, $p$-value = 0.01).

Statistically, $M_{IR}$ was related more to $HVI_{str}$ ($r = 0.46$, $p$-value = 0.008) than to $STE_{ten}$ ($r = -0.01$, $p$-value > 0.05) for all Upland datasets, echoing a previous inconclusive observation between $STE_{ten}$ and $M_{IR}$ on combined Upland fibers from the U.S. and four foreign countries [11]. A recent investigation reported that both MIC and maturity values were positively and significantly correlated with single fiber breaking force but negatively with single fiber strength on a set of Upland kinds of cotton constructed from a genetic approach [12]. Meanwhile, fiber $CI_{IR}$ was related less to $HVI_{str}$ ($r = 0.20$) than to $STE_{ten}$ ($r = -0.40$) for this Upland dataset. The observation agreed with a previous study that single fiber breaking tenacities against fiber XRD crystallinity differed between developing SJ-2 and Maxxa cotton fibers [3].

Notably, in Table 7, there was a weak correlation between either $M_{IR}$ and $STE_{elo}$ ($r = -0.01$, $p$-value > 0.05) or between $CI_{IR}$ and $STE_{elo}$ ($r = -0.22$, $p$-value > 0.05) for all Upland datasets and also for Pima fibers ($r = 0.10$ and $-0.14$, respectively). In general, three of six Upland cultivars (A, B, and E) showed an increase in $STE_{elo}$ with $M_{IR}$ ($r = 0.33$ to 0.99), whereas the other three cultivars indicated the opposite change ($r = -0.62$ to $-0.49$);

however, only Upland E fibers had a strong and significant correlation between the two ($r = 0.99$, *p*-value = 0.03). Unlike Upland B, C, D, and F cultivars, Upland A and E fibers were found to increase in $STE_{elo}$ with fiber $CI_{IR}$ ($r = 0.41$ to 0.99), in which only Upland A cultivar was to have a strong and significant correlation ($r = 0.99$, *p*-value = 0.01) among six Upland cultivars.

## 4. Conclusions

This study linked fiber Stelometer and HVI properties to fiber crystallinity and maturity indices determined by analyzing ATR FT-IR spectra of tiny Stelometer breakage specimens, aiming to examine the relationships between fiber physical and structure properties among six Upland and one Pima cotton cultivar. Although the plot of $STE_{ten}$ against $HVI_{str}$ implied a reasonable agreement between two strength measurements ($r = 0.57$ ***), the Upland D cultivar (immature with the smallest MIC, $M_{IR}$, and $CI_{IR}$ values) was observed to show the lowest $HVI_{str}$ value (=25.7) but a relatively larger $STE_{ten}$ value (=21.3). In contrast, the Upland F cultivar (having the greater $CI_{IR}$ index) showed the smallest $STE_{ten}$ value (=19.9) but a relatively larger $HVI_{str}$ value (=29.5).

A comprehensive examination of multiple relationships on individual Upland cultivars suggested strong and significant correlations between $CI_{IR}$ and $STE_{elo}$ for the Upland A fibers ($r = 0.99$ *), between $M_{IR}$ and $STE_{ten}$ for the Upland C fibers ($r = 0.90$ *), between $M_{IR}$ and $HVI_{str}$ for the Upland D fibers ($r = 0.98$ **), and also between $M_{IR}$ and MIC ($r = 0.99$*), as well as between $M_{IR}$ and $STE_{elo}$ ($r = 0.99$ *) for the Upland E fibers. Relatively, there existed a moderate and significant correlation between $CI_{IR}$ and $STE_{ten}$ for Pima fibers ($r = 0.58$ *). Different responses underscored the importance of experimental design and data analysis in understanding the unique response among any pair of fiber MIC, $HVI_{str}$, $STE_{ten}$, $STE_{elo}$, $M_{IR}$, and $CI_{IR}$ values, even within one cultivar. Beyond the approach reported here, other strategies might be explored with the ultimate purpose of improving fiber strength or elongation measurement, unraveling fiber strength or elongation's response to fiber structure, and enhancing fiber utilization and processing efficiency.

**Funding:** This research received no external funding.

**Institutional Review Board Statement:** Not applicable.

**Data Availability Statement:** Data are contained within the article.

**Acknowledgments:** The author sincerely thanks J. Linda and N. Carroll of ARS USDA for their technical assistance in conducting the Stelometer measurement. Mention of a product or specific equipment does not constitute a guarantee or warranty by the U.S. Department of Agriculture and does not imply its approval to the exclusion of other products that may also be suitable.

**Conflicts of Interest:** The author declares no conflicts of interest. Although the author is employed by the funding organization, the funders had no role in the design of the study, in the collection, analysis, or interpretation of data, or in the writing of the manuscript but approved the decision to publish the results.

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
