# Peer review of "Cotton Fiber Strength Measurement and Its Relation to Structural Properties from Fourier Transform Infrared Spectroscopic Characterization"

_textiles, doi:10.3390/textiles4010009_

Round 1

Reviewer 1 Report

Comments and Suggestions for Authors

Lui's manuscript is an extension of the research that the author has been doing for the last decade.  It attempts to correlate the mechanical properties of cotton fibres with the spectroscopic measurements. 

I would recommend acceptance of the manuscript after the following comments are corrected.

1. FTIR is known to assess the top layer of the fibre up to micrometres, whereas the bulk structure is essential for mechanical properties. Even the fibre breakage specimens have been measured, a more detailed explanation of how the measurements (tomography) have been made is desirable.

2. It is recommended to provide the raw data of the collected FTIR spectra and the details of the evaluation process in the supporting information.

3. SEM images of the fracture end of each fibre type may help to illustrate the differences in mechanical performance of the fibres.

4. The quality of the images must be improved.

Comments on the Quality of English Language

Even the abstract contains some language imperfections.

Reviewer 2 Report

Comments and Suggestions for Authors

1- Introduction, lines 39 to 40 are not clear and need to be rewritten.

2- Considering that a certain type of cotton is used in this study, is it possible to use the results for other fibers as well? Is the initial washing of the fibers effective on the final result?

3- The quality of the figures is very low and needs to be improved.

4- The FTIR technique is sensitive to changes in functional groups. How has the effect of impurities, especially surface impurities, been considered and measured?

5- Please add some results to the conclusion and correct this section.

Round 2

Reviewer 1 Report

Comments and Suggestions for Authors

In response to one of the comments the author replied: "The author will provide the collected FTIR spectra upon the request". As a reviewer I am requesting to provide the collected raw data FTIR spectra before I make a final decision.

Comments on the Quality of English Language

Minor language imperfections are tolerable

Reviewer 2 Report

Comments and Suggestions for Authors

Accept

Author Response

Thank you for the consideration.

Round 3

Reviewer 1 Report

Comments and Suggestions for Authors

I have no further comments

Comments on the Quality of English Language

ok